# Analysis of Affecting Factors of the Fate of Medial Meniscus Posterior Root Tear Based on Treatment Strategies

**DOI:** 10.3390/jcm10040557

**Published:** 2021-02-03

**Authors:** Jae Ik Lee, Dong Hyun Kim, Han Gyeol Choi, Tae Woo Kim, Yong Seuk Lee

**Affiliations:** 1Department of Orthopedic Surgery, Seoul National University College of Medicine, Seoul National University Bundang Hospital, Seoul 13620, Korea; jaeik15@gmail.com (J.I.L.); osdrkdh@gmail.com (D.H.K.); meinmed87@naver.com (H.G.C.); 2Department of Orthopedic Surgery, Seoul National University College of Medicine, Seoul National University Boramae Medical Center, Seoul 07061, Korea; orthopassion@naver.com

**Keywords:** knee, meniscal injury, osteoarthritis, treatment

## Abstract

Meniscal tear is a common reason for patients to undergo knee operation, and the medial meniscus posterior root tear (MMPRT) is one of the most frequent kinds of meniscal tears. The purpose of this study was to analyze participants’ factors (anthropometric and medical) to the fate of the MMPRT based on the treatment strategy. The hypothesis of this study was that treatment modalities from conservative treatment to final arthroplasty would be affected by participants’ affecting factors. From July 2003 to May 2018, 640 participants were included. Groups were categorized according to the treatment strategies such as conservative treatment, arthroscopic surgery, high tibial osteotomy (HTO) and arthroplasty surgery. Participants’ affecting factors were analyzed by one-way analysis of variance according to the four different treatment strategies and a correlation between affecting factors was also analyzed. Participants with K-L (Kellgren–Lawrence) Grade 4 and high BMI > 28.17 were appropriate candidates for arthroplasty, with K-L Grade 4 being a greater determining factor than high BMI. Participants with alignment factors such as low initial weight bearing line (WBL) (26.5%) and high delta WBL ratio (5.9%) were appropriate candidates for HTO, with the delta WBL ratio being a greater determining factor than initial WBL. Longer MRI-event times (1.44 year) and a lesser extent of meniscal extrusion (2.98 mm) were significantly associated with conservative treatment. Understanding the correlation of each affecting factor to the treatment strategy will help clinicians decide on the appropriate treatment for patients with MMPRT.

## 1. Introduction

The medial meniscus, a crescent-shaped fibrocartilaginous structure, is known to be essential for congruity, stability, shock absorption, and proprioception of the knee joint [1,2,3]. Furthermore, the medial and lateral meniscus maintain up to 90% and 70% of the knee’s hoop tension, respectively, which allows for accurate intra-articular load transfer [4]. Hoop tensions are generated as axial forces and converted to tensile stresses along the circumferential collagen fibers of the meniscus. However, radial tear of the posterior meniscal root can lead to a loss of circumferential hoop tension. The loss of hoop tension increases the contact pressure and accelerates the progression of osteoarthritis (OA) [5,6]. OA is a type of joint disease that results from breakdown of joint cartilage and underlying bone. Radial tears of the medial meniscus posterior horn are very common among Asian individuals, and the majority of these tears are expressed as a medial meniscus posterior root tear (MMPRT) [7]. MMPRT is defined as either a radial tear located within 1 cm of the meniscal attachment or a bony root avulsion [8]. Root tears are more frequently observed in the medial meniscus than in the lateral meniscus because the posterior horn is less mobile than lateral meniscus [9].

MMPRT has recently received greater attention because it has been associated with the development of excessive meniscal extrusion and accelerated degenerative changes [9]. An important characteristic of MMPRT is that it occurs during the early stages of OA that may be reversible for the OA progression. In addition, the incidence of MMPRT is higher than expected. The probability of the MMPRT in OA participants is approximately 80%, and MMPRT accounts for 27.8% of all medial meniscal tears [7]. Therefore, it may be important to identify which participants are likely to be managed well with conservative treatments such as medication or injection therapy and which participants should be referred for surgical treatment. However, there is little consensus regarding this topic, and current treatment strategies, including conservative treatment, meniscectomy or meniscal repair, realignment procedures such as a high tibial osteotomy (HTO), and even arthroplasty, are highly variable.

Another important characteristic of MMPRT is that it is commonly observed with degeneration rather than with acute traumatic injury, which mostly occurs in other tear types. Therefore, it may be important to predict the course of MMPRT and identify which affecting factors are related to specific treatments. This study aimed to analyze the participants’ affecting factors of the fate of MMPRT based on treatment strategies. The hypothesis of this study was that treatment modalities from conservative treatment to final arthroplasty would be affected by participants’ affecting factors.

## 2. Materials and Methods

Six hundred and forty patients who were diagnosed with MMPRTs from July 2003 to May 2018 were included in this retrospective cohort study. The mean age of the participants was 58.09 (range: 21–86) years, and the average follow-up period was 3.75 (range: 2–11) years. MMPRTs were diagnosed on the basis of the findings of magnetic resonance imaging (MRI) or data on outpatient chart review conducted by the Clinical Data Warehouse (CDW) in Seoul National University Bundang Hospital.

All participants were divided into four groups based on the final treatment: arthroplasty, high tibial osteotomy (HTO), arthroscopic surgery, and conservative treatment. The exclusion criteria were as follows: (1) a history of trauma such as a periarticular fracture or ligament injury; (2) infection; (3) inflammatory arthritis; (4) previous meniscal injury and/or knee operation. Institutional Review Board approval was obtained before the commencement of this retrospective analysis.

### 2.1. Extraction of Affecting Factors

All the participants’ affecting factors selected for this study were surveyed from previous articles [10,11,12,13]. Possible affecting factors of operation or OA progression were gender, age, body mass index (BMI), duration of attempted conservative treatment, malalignment as a weight-bearing line (WBL) ratio, proximal tibial morphology, Kellgren–Lawrence grading scale (K-L grade), bone marrow lesions (BMLs), and severity of meniscal extrusion. The affecting factors were categorized as anthropometric or medical factors.

#### 2.1.1. Anthropometric Factors

The anthropometric factors included age, gender, BMI, and interval of conservative treatment. BMI was defined as the participant’s weight in kilograms divided by body height (in meters squared). The interval of conservative treatment was expressed as ‘MRI-event time.’ MRI was described as to the time when MRI was obtained and event as the date of surgery or the last visit of the outpatient department. The period between MRI and event was defined as the MRI-event time.

#### 2.1.2. Medical Factors

Standing knee anterior-posterior, lateral, and hip-knee-ankle views were evaluated in routine follow-ups, and coronal, sagittal MRIs were evaluated in all participants. The WBL ratio, proximal tibial morphology, K-L grade, BMLs, and severity of meniscal extrusion were evaluated. The mechanical axis deviation was evaluated using the initial WBL and delta WBL ratios. Proximal tibial morphology was evaluated using tibial varus angle (TVA) and posterior tibial slope (PTS). OA severity was evaluated using the K-L grade. BMLs were evaluated using the MRI osteoarthritis knee score (MOAKS). Meniscal extrusion was evaluated on the coronal MRI. INFINITT (ver. 5.0.9.2, Seoul, Korea) was used for all radiographic assessments and measurements.

##### WBL

The WBL ratio was calculated by measuring the distance from the medial edge of the proximal tibia to the point where the WBL intersected the proximal tibia and by dividing the measurement by the entire width of the tibia. A percentage was calculated by multiplying this ratio by 100%. The delta WBL ratio was calculated as the difference between the initial WBL ratio and the WBL ratio just before operation or the last follow-up in the outpatient department for the participants who did not undergo operation.

##### TVA

The TVA was defined as the angle between the line perpendicular to the tibial shaft and the articular surface of the proximal tibia [14].

##### PTS

The PTS was defined as the angle between the line connecting the highest anterior and posterior points of the medial plateau and the line perpendicular to the anterior tibial cortex [10].

##### K-L Grade

The K-L grade was determined using anteroposterior knee radiographs. Each radiograph was assigned a grade from 0 to 4 based on the extent to which it correlated with an increasing OA severity. Grade 0 indicates no presence of OA and Grade 4 indicates severe OA [15].

##### MOAKS

The MOAKS instrument refined the scoring of bone marrow lesions (BMLs) [16] (Figure 1).

##### Meniscal Extrusion

The extent of medial meniscal extrusion was measured from the medial margin of the tibial plateau to the medial margin of the medial meniscus on the image at the midpoint of the femoral condyle. Osteophytes were excluded for the determination of the margin of the tibial plateau [10] (Figure 2).

### 2.2. Statistical Analysis

All parameters were expressed as mean plus-minus standard deviations. Categorical variables were analyzed using one-way analysis of variance (ANOVA) and Duncan’s post-hoc analysis, Pearson correlation analysis, and linear regression analysis. Receiver operating characteristic (ROC) curves were generated and used to determine the best cutoff value for each affecting factor. A ROC curve is a graphical plot that illustrates the diagnostic ability of a binary classifier system as its discrimination threshold varies. Results were considered statistically significant when the *p*-value was <0.05 [17]. Level of significance was set a priori at *p* < 0.05. The kappa coefficient was used to evaluate the reliability of radiographic evaluations. The kappa values ranged from 0 to 1, with 1 indicating perfect agreement between two observers. All statistical analyses were performed using the Statistical Package for the Social Sciences (version 22.0, IBM, Armonk, NY, USA).

## 3. Results

The inter- (kappa = 0.816) and intra-observer (kappa = 0.853) reliabilities were acceptable. The detailed anthropometric and medical data are listed in Table 1. Regarding the male/female ratio, there was no significant difference among the groups (*p* = 0.393). In terms of the mean age, arthroscopic treatment was preferred for participants of younger age, and HTO and conservative treatments were preferred for participants with similar mid-ages. Arthroplasty was preferred for participants of older age (*p* = 0.001). BMI was the highest in the arthroplasty group (*p* = 0.001). Participants in the arthroplasty group had significantly higher BMI than those in the HTO, arthroscopic treatment, and conservative treatment groups (*p* = 0.020, 0.001, and 0.004, respectively). The MRI-Event time was the longest in the conservative treatment group (*p* = 0.001). The arthroplasty, HTO, and arthroscopic treatment groups showed significantly shorter MRI-Event times than the conservative treatment group (*p* = 0.001, 0.003, and 0.010, respectively) (Figure 3).

The WBL ratio was the lowest in the HTO group (*p* = 0.001). The HTO group had a significantly lower initial WBL ratio than the arthroplasty, arthroscopic treatment, and conservative treatment groups (*p* = 0.011, 0.018, and 0.009, respectively). The HTO group had a significantly higher delta WBL ratio than the arthroplasty, arthroscopic treatment, and conservative treatment groups (*p* = 0.021, 0.030, and 0.018, respectively). Regarding the parameters of proximal tibial morphology, the TVA and PTS were not significantly different among the four groups (*p* = 0.218 and 0.377, respectively). The K-L grade was the highest in the arthroplasty group (*p* = 0.001). The arthroplasty group had a significantly higher K-L grade than the HTO, arthroscopic treatment, and conservative treatment groups (*p* = 0.010, 0.031, and 0.013, respectively). Regarding the parameters of BMLs, the MOAKS was not significantly different among the four groups (*p* = 0.762). The extent of meniscal extrusion was the least in the conservative treatment group (*p* = 0.001). The conservative treatment group had significantly less meniscal extrusion than the arthroplasty, HTO, and arthroscopic treatment groups (*p* = 0.031, 0.029, and 0.035, respectively) (Figure 4). Regarding the comparison between the conservative and surgical treatment groups, the MRI-event time was significantly longer in the conservative treatment group than the surgical treatment groups (*p* = 0.032). Meniscal extrusion was significantly lesser in the conservative treatment group than the surgical treatment groups (*p* = 0.018). There were no significant differences in the other affecting factors (Figure 5).

In the correlation analysis, three pairs of affecting factors had a significant correlation. The multivariate Pearson correlation analysis revealed that BMI was significantly correlated with the K-L grade (r = 0.367). The initial WBL ratio was significantly correlated with the delta WBL ratio (r = −0.332). In addition, meniscal extrusion was significantly correlated with the MRI-event time (r = −0.418) (Table 2). Meanwhile, the linear regression analysis showed that BMI was significantly associated with the K-L grade (*p* = 0.021, ß = 0.492). The initial WBL ratio was significantly associated with the delta WBL ratio (*p* = 0.034, ß = −0.195). In addition, meniscal extrusion was significantly associated with the MRI-event time (*p* = 0.029, ß = −0.924) (Table 3). In the analysis of significant independent variables, BMI was the highest in the arthroplasty group, the initial WBL ratio was the lowest in the HTO group, and the extent of meniscal extrusion was the least in the conservative treatment group (Figure 6).

The ROC curve analyses were conducted to determine the cutoff values for each affecting factor, including BMI, the K-L grade, the initial WBL ratio, the delta WBL ratio, meniscal extrusion and the MRI-event time. The cutoff values for BMI and the K-L grade for arthroplasty were >28.17 and 3, respectively (Figure 7A). The cutoff values for the initial WBL and delta WBL ratio for HTO were ≤26.5 and >5.9, respectively (Figure 7B). The cutoff values for meniscal extrusion and the MRI-event time for operation were >2.98 and ≤1.44, respectively (Figure 7C). Two parameters were related to the different treatment strategies. In the comparison between those two parameters, the K-L grade, delta WBL ratio, and the MRI-event had greater separability than BMI, the initial WBL ratio, and meniscal extrusion in the arthroplasty, HTO, and conservative treatment groups, respectively (Table 4).

## 4. Discussion

The principal finding of this study was the identification of important factors that affect the fate of the MMPRT, based on treatment strategies. Based on the results, our study suggests a prediction model for the selection of specific treatment strategies. Most previous studies have assessed only one or two factors; conversely, in this study, we performed a multifactorial analysis of relevant affecting factors according to four different treatment strategies and analyzed the correlations among the affecting factors. We also provide a specific cutoff value for each factor. Based on the study results, patients with K-L Grade 4 and high BMI (>28.17 kg/m^2^) could be appropriate candidates for arthroplasty, with K-L Grade 4 being a greater determining factor than a high BMI. Meanwhile, those with alignment factors, such as a low initial WBL ratio (26.5%) and high delta WBL ratio (5.9%), could be appropriate candidates for HTO, with the delta WBL ratio being a greater determining factor than the initial WBL ratio. Longer MRI-Event times (1.44 year) and a lesser extent of meniscal extrusion (2.98 mm) were significantly associated with conservative treatment. Therefore, our hypothesis was verified.

MMPRTs are strongly associated with obesity and are usually accompanied by degenerative chondral damage of the involved compartment [18]. Similarly, another study demonstrated significant associations between increasing BMI and meniscal tears, leading to the need for surgical interventions [19]. In addition, positive correlations have been reported between BMI and the K-L grades [20]. We also identified that the participants with a high BMI developed OA and that the probability of arthroplasty was strongly related to this factor. Varus alignment is a strong risk factor for medial joint degeneration. Previous studies have reported that varus alignment is consistently identified as a strong predictor of medial knee OA progression in participants with MMPRT. In addition, HTO for participants with degenerative meniscal root tears may present an excellent opportunity for early intervention in participants with varus alignment and symptomatic early knee OA for secondary prevention of radiographic knee OA [21]. In our study, we found an association between a lower initial WBL ratio and a high probability of HTO. Furthermore, we found a negative correlation among affecting factors such as the initial WBL ratio and delta WBL ratio, which has not been previously reported. Meniscal extrusion can be regarded as the disruption of meniscal hoop tension. A previous study has reported that more than 3-mm medial meniscal extrusion was strongly associated with degenerative joint disease [22]. Another study also reported that meniscal extrusion assessment may be important for determining the optimal treatment strategy for MMPRT [11]. In our study, we found an association between larger meniscal extrusion and higher probability of surgical intervention irrespective of the specific operation. Furthermore, we found a negative correlation among affecting factors such as meniscal extrusion and the interval of conservative treatment, which has not been previously reported. Therefore, our second hypothesis was also verified.

Proximal tibial morphology is closely associated with MMPRT progression to OA. [22] Okazaki et al. [23] reported that a steep posterior slope is a risk factor for MMPRTs. Most researchers agree that there is a higher incidence of MMPRTs in participants with a higher PTS [24]. However, this study only reported on the relationship between proximal tibial morphology and MMPRT incidence, and no study has assessed further treatment strategies. In the current study, regarding the parameters of proximal tibial morphology, TVA and PTS were not shown to be statistically significantly correlated with each treatment strategy and were not associated with specific treatment strategies or a strong determining factor for operation. Another study reported that the likelihood of arthroplasty was higher among participants with BMLs [13]. Participants with bone marrow edema and OA have an increased risk for arthroplasty compared to those with OA without marrow edema [25]. However, contrary to results of previous studies, the results of our study were analyzed to be irrelevant. The reasons for this can be explained as follows: BMLs around the knee can be classified into traumatic or non-traumatic and into reversible or irreversible [26]. Previous studies have included older participants with irreversible BMLs, whereas our study included participants with traumatic and non-traumatic BMLs. For this reason, in the current study, regarding the parameters of BMLs, MOAKS was not related to each treatment strategy.

The strength of this study was that it is the first study to demonstrate an association between affecting factors and outcomes of MMPRTs with respect to the specific treatment strategies. The most important contributions of this study would be its assessment of MMPRT outcomes and the establishment of a prediction model for MMPRTs, which are important for early OA management. This study was also performed at a single center. Moreover, blinded analyses were performed by five knee specialists, and similar results were obtained when each surgeon analyzed the data individually.

### Limitations

This study also had some limitations. First, the selection of a specific operation could be dependent on a surgeon’s indication and preference. Therefore, the specific cutoff value and results could be different from our results. Second, although the presence of MMPRT was determined in accordance with a standardized definition, it was difficult to avoid the influence of subjective judgment because confounding factors such as the presence of osteophytes were expressed in measuring the degree of extrusion. Third, there was still the possibility of OA progression, and participants could subsequently be converted to another treatment strategy. However, this may be beyond the scope of this study, thus, we did not focus on secondary treatments.

## 5. Conclusions

A low initial WBL ratio was associated with a high possibility of realignment procedures, such as HTO; a high BMI was more likely to result in arthroplasty. A large meniscal extrusion was found to be an important risk factor for any kind of operative treatment. A low initial WBL ratio, high BMI, and large meniscal extrusion were correlated with a higher delta WBL ratio, higher K-L grade, and shorter interval of conservative treatment, respectively. Understanding the correlation of each affecting factor to the treatment strategy will help clinicians decide on the appropriate treatment for patients with MMPRT.

## Figures and Tables

**Figure 1 jcm-10-00557-f001:**
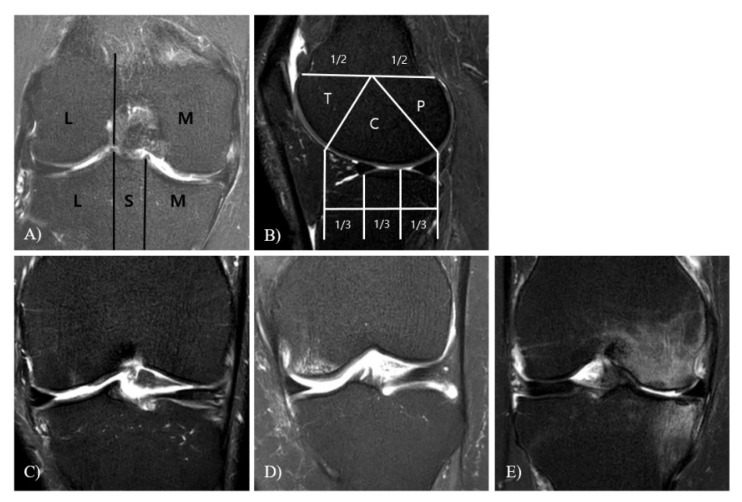
The scoring system magnetic resonance imaging (MRI) osteoarthritis knee score (MOAKS) for assessing bone marrow lesions (BMLs). (**A**) Coronal image showing anatomical delineation of the tibia into the medial, sub-spinous (SS), and lateral subregions. The femur was divided into the medial and lateral femoral condylar regions. The intercondylar notch was part of the medial femur. (**B**) Anatomical delineation of the femur into the trochlear (T), central (C), and posterior (P) regions on sagittal projection. Sagittal projection depicts delineation of the tibia into the anterior, central, and posterior subregions, which was divided into equal thirds. (**C**–**E**) BML grading. Grade 0 = none, Grade 1 < 33% of the sub-regional volume, Grade 2 = 33–66% of the sub-regional volume, and Grade 3 > 66% of the sub-regional volume. (**C**) Coronal T2-weighed image showing small Grade 1 BML in the central subregion of the medial femoral condyle. (**D**) A Grade 2 BML is depicted in the central subregion of the medial femur. (**E**) Grade 3 BMLs are observed s in the central subregions of the medial femur and central medial tibia.

**Figure 2 jcm-10-00557-f002:**
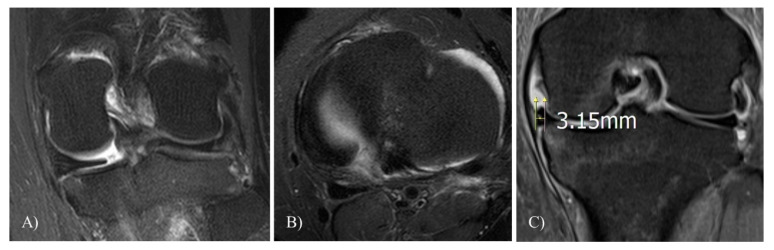
Measurements of extent of the medial meniscal extrusion on magnetic resonance images. (**A**,**B**) Meniscus posterior root tear (MMPRT) is prominently observed. (**C**) The extent of medial meniscal extrusion was measured from the medial margin of the tibial plateau to the medial margin of the medial meniscus on the image at the midpoint of the medial femoral condyle.

**Figure 3 jcm-10-00557-f003:**
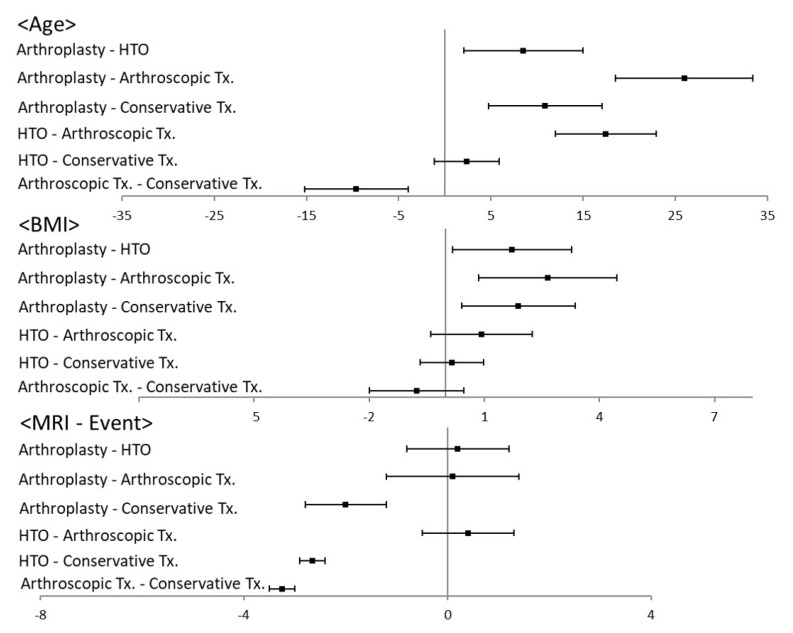
Results of post-hoc analysis for anthropometric affecting factors.

**Figure 4 jcm-10-00557-f004:**
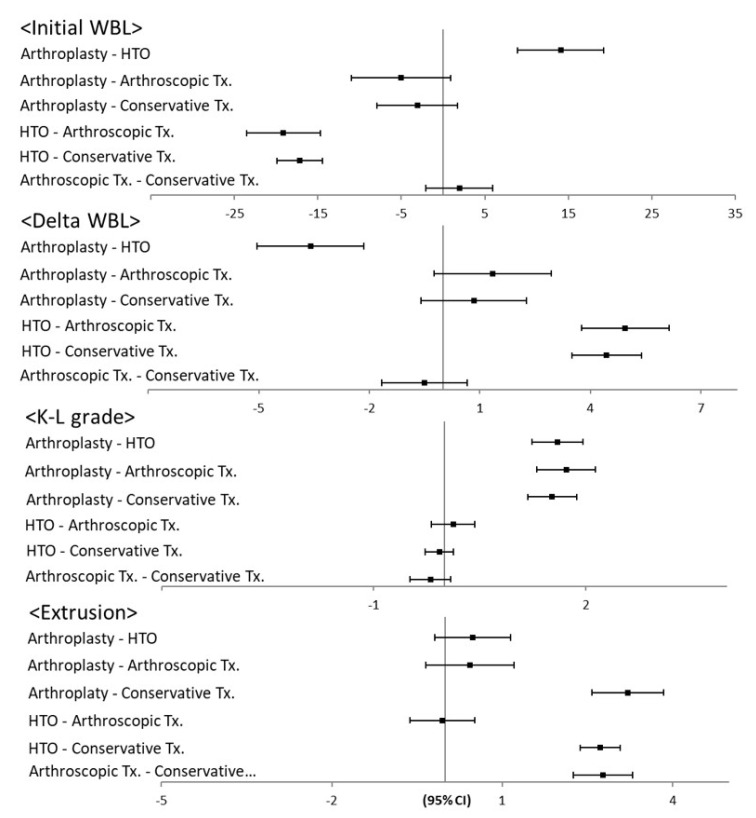
Results of post-hoc analysis for medical affecting factors.

**Figure 5 jcm-10-00557-f005:**
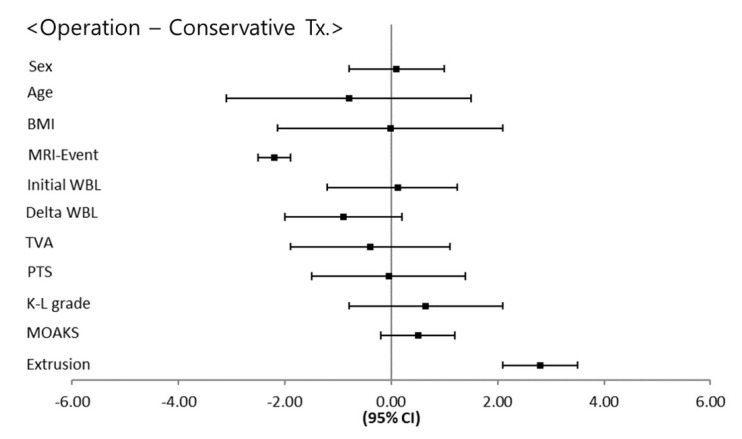
Results of post-hoc analysis for surgical versus conservative treatment.

**Figure 6 jcm-10-00557-f006:**
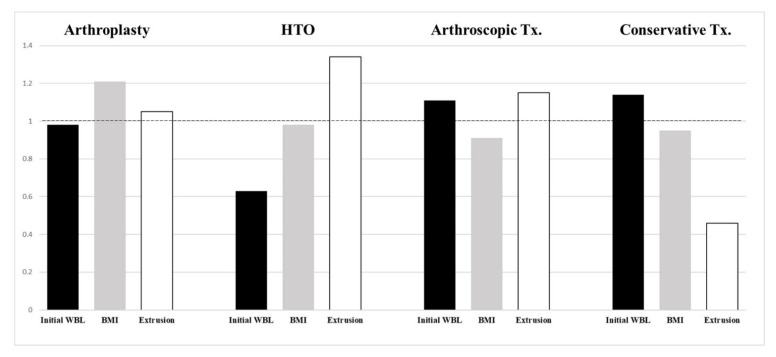
Analysis of significant independent variables. The mean value of all factors was set to 1 and compared using the ratio. BMI was the highest in the arthroplasty group, the initial WBL ratio was the lowest in the HTO group, and meniscal extrusion was the least in the conservative treatment group.

**Figure 7 jcm-10-00557-f007:**
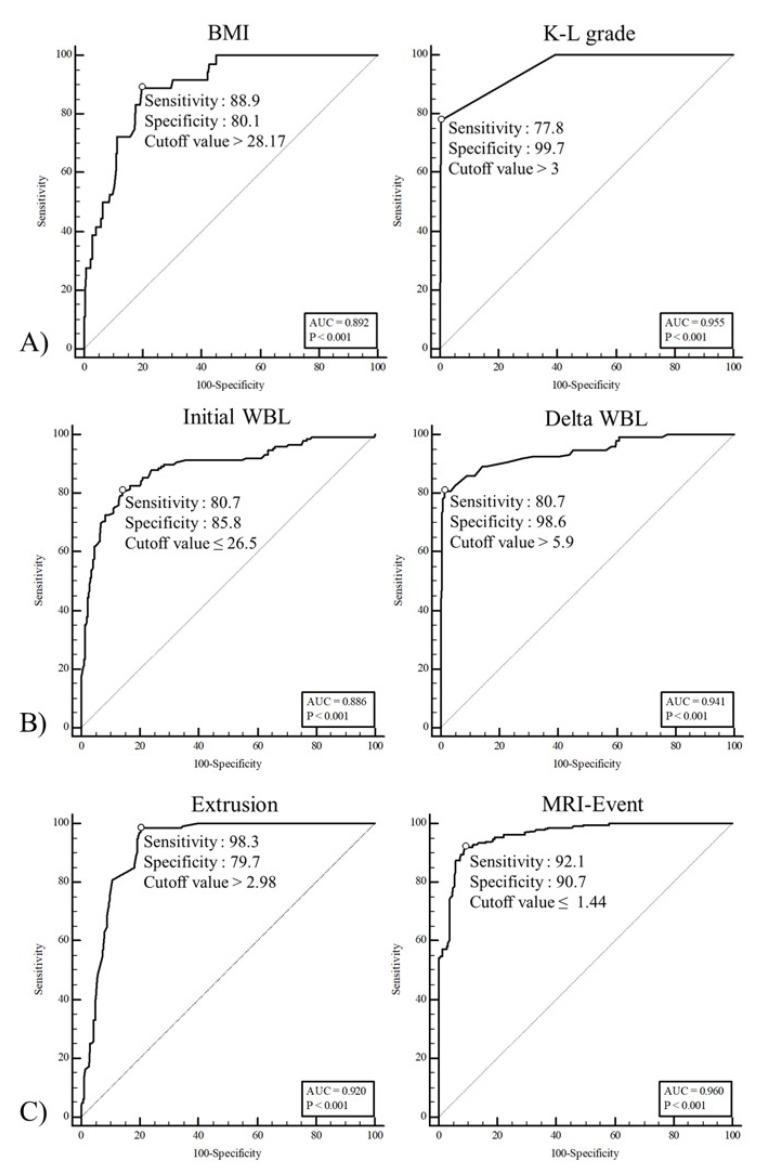
Receiver operating characteristics (ROC) curves for the associations of (**A**) arthroplasty with BMI and the Kellgren–Lawrence (K-L) grade, (**B**) high tibial osteotomy (HTO) with the initial WBL and delta WBL ratios, and (**C**) operation with extrusion and the MRI-event.

**Table 1 jcm-10-00557-t001:** Anthropometric and medical factors in each groups.

	Arthroplasty (36)	HTO (150)	Arthroscopic Tx. (55)	Conservative Tx. (399)	*p*-Value
Male/Female ratio	5:31	31:119	24:31	155:244	n.s.
**Anthropometric**					
Age (years)	69.25 ± 6.02	58.12 ± 4.18	43.26 ± 15.26	57.13 ± 15.16	0.001 *
BMI (kg/m²)	29.72 ± 2.83	26.67 ± 3.16	26.42 ± 2.63	25.92 ± 2.87	0.001 *
MRI—Event (years)	1.17 ± 0.97	0.88 ± 1.32	0.95 ± 0.41	3.13 ± 2.21	0.001 *
**Medical factors**					
Initial WBL (ratio)	31.44 ± 16.42	21.06 ± 11.03	38.22 ± 10.59	36.24 ± 9.32	0.001 *
Delta WBL (ratio)	4.21 ± 1.77	7.32 ± 3.46	3.06 ± 2.10	2.76 ± 1.09	0.001 *
TVA (degree)	5.32 ± 1.17	5.44 ± 0.85	5.18 ± 0.84	5.07 ± 1.60	n.s.
PTS (degree)	11.25 ± 2.55	11.03 ± 1.72	10.85 ± 1.44	11.11 ± 3.48	n.s.
K-L grade	3.78 ± 0.42	2.18 ± 0.39	2.05 ± 0.65	2.01 ± 1.06	0.001 *
MOAKS	2.11 ± 0.67	1.96 ± 0.69	1.98 ± 0.71	1.83 ± 1.02	n.s.
Extrusion (mm)	3.13 ± 1.36	3.98 ± 1.01	3.43 ± 1.09	1.36 ± 1.51	0.001 *

Values are presented as mean ± standard deviation. WBL: weight bearing line; BMI: body mass index; TVA: tibia varus angle; PTS: posterior tibial slope; MOAKS: MRI osteoArthritis knee score. * Denotes statistically significant.

**Table 2 jcm-10-00557-t002:** Correlation coefficients between affecting factors.

Variable	Sex	Age	BMI	MRI-Event	Initial WBL	Delta WBL	TVA	PTS	K-L Grade	MOAKS	Extrusion
**Sex**	1	0.091	0.121	0.131	0.089	0.187	0.023	0.119	0.087	0.076	0.107
**Age**		1	−0.008	0.093	0.121	0.129	−0.019	0.110	0.251	0.095	0.117
**BMI**			1	−0.217	0.093	0.212	0.091	0.034	0.367 *	0.198	0.201
**MRI-Event**				1	0.132	−0.229	0.079	0.101	0.211	−0.103	−0.418 *
**Initial WBL**					1	−0.332 *	0.013	0.094	0.202	0.081	0.071
**Delta WBL**						1	0.113	0.019	0.204	0.109	0.208
**TVA**							1	0.131	0.042	0.077	0.016
**PTS**								1	0.076	0.109	0.129
**K-L Grade**									1	0.098	0.142
**MOAKS**										1	0.173
**Extrusion**											1

Data are presented as correlation coefficients. The statistical significance was set at *p* < 0.05. * *p*-value < 0.05. WBL: weight bearing line; BMI: body mass index; TVA: tibia varus angle; PTS: posterior tibial slope; MOAKS: MRI osteoarthritis knee score; PCC: Pearson correlation coefficient. * Denotes statistically significant.

**Table 3 jcm-10-00557-t003:** Subgroup analysis using a linear regression analysis.

Independent Variable	Dependent Variable	Regression Coefficient (ß)	*p* Value
Initial WBL	Delta WBL	−0.195	0.034 *
BMI	K-L grade	0.492	0.021 *
Meniscal extrusion	MRI-Event	−0.924	0.029 *

The statistical significance was set at *p* < 0.05. * *p*-value < 0.05 WBL: weight bearing line; BMI: body mass index.

**Table 4 jcm-10-00557-t004:** ROC curve analysis.

Variable	AUC	SE	*p* Value	95% CI
BMI	0.892	0.021	0.001 *	0.865–0.915
K-L grade	0.955	0.014	0.001 *	0.935– 0.969
Initial WBL ratio	0.886	0.017	0.001 *	0.859–0.910
Delta WBL ratio	0.941	0.013	0.001 *	0.919–0.958
Extrusion	0.920	0.011	0.001 *	0.896–0.940
MRI-event	0.960	0.007	0.001 *	0.942–0.974

The statistical significance was set at *p* < 0.05. * *p*-value < 0.05. AUC: area under cover; SE: standard error; CI: confidence interval; WBL: weight bearing line; BMI: body mass index.

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
