# Peer review of "Analysis of Affecting Factors of the Fate of Medial Meniscus Posterior Root Tear Based on Treatment Strategies"

_jcm, 2021, doi:10.3390/jcm10040557_

Round 1

Reviewer 1 Report

Analysis of Affecting Factors of the Fate of Medial Meniscus 2 Posterior Root Tear Based on Treatment Strategies

Overview: The aim of this study was to analyze affecting factors affecting Medial Meniscus 2 Posterior Root Tear (MMPRT) based on the treatment strategy. This is an interesting study that reports important and useful information. However, the quality of the writing and composition is very poor, which detracts from understanding the material presented. Therefore significant changes must be made to the text to improve the flow of the manuscript. Some suggestions are provided below. 

Major comments:

Title: the title should include the keyword osteoarthritis.

Abstract: the scientific English language used in the abstract must be improved. Also, the abstract must include one or two sentences to provide the necessary background information as well as a clear sentence at the very end, summarising the conclusions. The abstract is missing these components.

Keywords: medial meniscus is missing from the keywords.

The scientific English language in the manuscript must be improved by using the services of a manuscript editing service. There are too many errors to list in this review and it is not the responsibility of the peer reviewer to provide a line by line set of corrections for the authors. The authors really should have prepared a more professionally written manuscript before submission. 

However, to help the authors a few examples are provided below:

Discussion: in the first sentence of this section the word principle is used in the incorrect context. The sentence should read: The principal finding of this study was identification of important factors that affect the fate of the MMPRT, based on treatment strategies employed. 

Reviewer 2 Report

Thank you for the opportunity to review this manuscript. The purpose was the investigation of the affecting factors of medial meniscus root tear based on various treatment strategy. I completely see the interest of the topic for the readership of the scope of Clinical Medicine. However, the clarity of writing should be strongly improved and I have additional suggestions that would improve the manuscript.

Major general comments:

Native speaker revision of the manuscript should be taken into account by the authors.

Abstract:

In general, you have way too many abbreviations used in the Abstract. You should not use them without first definition in the Abstract as well. Please revise the whole Abstract according to that. You can use the abbreviations when they are commonly used, however, I don’t see that these are commonly used and widely known abbreviations. It makes the Abstract very hard to follow and understand, because the reader just does not know what some abbreviations mean.

L10: Avoid abbreviations in the Abstract without introducing them at first appearance.

L11: Avoid the term patient, use “participant” instead throughout the manuscript.

L11/12: Rephrase this sentence. It is quite misleading how it’s stated right now. E.g., “Groups were categorized according to the treatment strategies…”

L12/13: I’m struggling with the understanding of the sentence.  How was composition analyzed and why did you calculate correlations between the treatment groups. You should clarify the objective of your study also in the Abstract.

L14 et seq.: Reduce the amount of statistical metrics in the Abstract.

L14 et seq.: Beforehand you wrote that you analyzed the composition of the treatment strategies and the correlations of them. However, in this section you’re reporting only the significant differences! By what statistical computation did you obtain them. What was the objective for that. You have to include the study’s objective / research questions in the Abstract as well. Otherwise it is misleading content.

L20/21: You should also add a conclusive sentence at the end of the Abstract. What do your results mean for practical implications?

L22 Keywords: You chose quite weak keywords (e.g., knee, osteoarthritis, affecting factors). Try to find more powerful ones (E.g., Meniscal root tear treatment). Think about the quality of the keywords, because they highly enhance the visibility of your manuscript.

Manuscript:

Introduction:

General comment:

Generally, the introduction covers all necessary content to clearly deduce your research questions. However, I have to argue that your sometimes too vague in your statement. Please, try to be more precise towards the research question you’re addressing.

Specific Comments:

L27: Is there a better terminology for “hoop tension”, as it is mainly used for machines and tools. Or could you clarify the meaning of this term briefly in the Introduction.

L27/28: There is a missing link between the two sentences. Why does it become necessary to apply the (surgical) treatment strategy in L28/29? Probably, the sentence L28-30 should come previously as it explains what consequences a root tear has.

L32-37: This is more general content than the latter part of the first section. You should include that before your specific explanation of treatment strategies. Build up your Introduction from general to specific study-related content.

L40/41: This sentence needs grammatical revision.

L42/43: I’m not fully convinced by tat statement. Is it just a two-way road? From the content before I assume there more strategies of treatment than just surgical operation and self-management. I would distinguish between surgical/invasive and conservative treatment strategies.

L50/51: Your hypotheses are very vague and undirected. If stating hypotheses instead of open research questions at the end of the Introduction, please, imply a clearer direction of the hypotheses – what you’ve expected beforehand? Subsequently, the aim would be to verify or falsify these hypotheses.

Methods:

General comment:

I do not see the reason why starting the Methods section with App development. Please firstly describe your sample, followed by the testing procedure alongside with the required technical developments for your study. Finalize with the data analysis and / or statistics.

Specific comments:

L53: Needs to be introduced at first appearance.

L53/54: Sentence needs grammatical revision. Or make two sentences out of it for better understanding.

L54: Please report here also the full affiliation of the institution (Clinical Data Warehouse or your clinic).

L55: Please give information about the number of participants in each sub-group. If not in the text, provide a Table with your whole sample or each sub-group’s anthropometric characteristics.

L57/58: Accordingly, did you acquire written informed consent for study participation of the participants. This is mandatory to report. Please, refer to the Helsinki declaration on ethical guidelines for medical research.

L64: I rather recommend the differentiation into anthropometric/individual and medical factors.

L65: If building sub-section according to 2.1, I would recommend to sub-order 2.2 and 2.3 as 2.1.1 and 2.1.2

L67: “divided by body height (in meters squared)”

L67/68: I do not understand the content and meaning of the sentence. Please revise.

Et seq.: I recommend to rewrite this section (now 2.2).

L72: If reporting MRI or radiologic analysis please provide the planes you have analyzed (i.e., sagittal etc.).

L73: Here you’re introducing the abbreviation of MRI after using it already several times beforehand. Please correct.

L78: But the version into the parentheses.

L80 et seq.: Revise the whole section for better understanding. Maybe it’s better to list all the affecting radiologic factors instead of writing the down in plain text. Furthermore, each affected factor should be directly linked to sub-figure of Figure 1A and 1B and not just referencing after the whole section, see Figure 1A. Figure 1A has five sub-figures and I, as a reader, do not want to puzzle out, which affecting factor belongs to which figure.

L94 et seq.: Some of the content of these very profound figure captions should be added into the section where it belongs to. Keep the captions clear, short and brisk and stay to the guidelines of the Journal: https://www.mdpi.com/journal/jcm/instructions#figures.

L111: Avoid mathematical symbols in the text.

L111/112: Please be more precise, which analysis of variance (ANOVA) you did apply. Once again, here you reporting that you’ve also computed ANOVAs, that is mandatory to report in the Abstract as well. Also how did you handle your statistical analyses after obtaining statistical differences in the ANOVA – did you also computed post hoc t-tests subsequently? This is necessitated to identify to clearly find which groups clearly differ and which not.

L113 et seq.: You have to report references for the statistical measures you have applied (especially, for the applied statistical computations, threshold p- and kappa-values). Start sentences with capital letters.

Results:

General comment:

Please, provide more structure in reporting your results. It’s a continuous forth and back of demographic and radiologic factors. Please report the results as you listed them in Table 1. So one can follow the Table and the text. Furthermore, maybe listing the results instead of prosaic writing would help to follow your results better.

L118 et seq.: Here you have the nicely depicted summary of your sample characteristics. I’d rather like to have this in the Method section.

Table 1: You give ratios of male-females. However afterwards you report means of the whole group. If gender-separating you have to separate their values through all variables.

Furthermore, what does the p-value tell? You found a significant difference over all groups? Fine but we should not which of the four groups exactly differ from each other and which not. This is valid for all reported variables. It clearly appears by the means and standard deviations that some values differ from each other and some not.

L149-154: These graphs need to get fully revised. The quality of the figures is pretty low and appears to be a screenshot or low-resolution pic. I recommend to include bigger and higher-resoluted figures. Additionally, check your figure caption. Official title is Figure 2, but from L152 onwards your report, 4A), 4B), and 4C) – I just don’t get this.

L155: Sentence needs complete grammatical revision.

L184: You should report the definition of the ROC curves exemplary in the Methods section.

L186-188: Check the editing of that lines.

Please, consider a new set up Results section. In the present structure it is pretty hard to follow and to see what your major and minor findings/results are.

Discussion:

General: At the beginning of the Discussion section, it is mandatory to report the objective of the study once again, before reporting the major findings followed by the report of minor findings. Best way to follow your thoughts, is to follow the structure of the results section.

The Discussion is generally all right. However, I suggest to clearly revising it in terms of a clearer structure of your findings and how they related or not to findings of the studies you refer to. This should be written more coherent. Right now, it’s rather a listing, which is hard to follow. I would rather like to read clear practical implications of your findings, as do’s or don’ts that we can deduct from your study, instead of just listing your results again.

L210-215: You should clearly discuss how you deduce the results into those findings. Why are some better candidates for HTO than others, for example, on which metric and which deduction do you base that assumption.

L226-228: Check the text editing please; line-space.

L248-252. This content could come at the very beginning of your Discussion, because it sums up what you’ve done in your study.

Limitations:

L257: I would see it a strength of your study instead of a limitation that all participants were treated in the same center. In such studies it is always important to control for such variations.

Conclusion:

Please, try to find clear conclusions of your results/findings. You’re again telling us plain results, especially at the end. What does a high correlation stand for? That would be of interest for the reader.

Round 2

Reviewer 1 Report

The authors have done a sub-standard and poor job of the revision. Many of the revisions proposed by this reviewer have not been considered. The revised manuscript requires substantial editing for correct use of scientific English language. There are incomprehensible sentences throughout the paper and numerous grammatical errors, words that have been spelt with capital letters that do not need to be spelt with capital letters and numerous other errors and inconsistencies.

The revised paper requires substantial further revisions.

Reviewer 2 Report

Thank you for the revised manuscript.
I highly appreciate your intense work in improving the quality of the article, which you surely achieved.

However, before accepting, I have a view lingering comments.

Abstract:

I appreciate the work you put into the Abstract and the readability and understanding was highly improved. However, in L12 and onwards you’re always using the unspecific term “affecting factors”. You should clearly specify in the Abstract at first mentioning, which your affecting factors have been.
L13/14: I see the improvement of adding your hypotheses to the Abstract. However, they are quite unspecific how they are stated at present. Please state your hypotheses more directed. Including dependent variables and not just affecting factors. Especially, hypothesis 1) is very open – you could put everything into it.
L24/25: This is not a real conclusion for me. It rather adds another result. Conclusion represents the summarized practical findings or implications of your results. This needs to be still adapted.

Keywords: You should not use keywords, which already appear in the title. I.e., Medial meniscal posterior horn root tear. Maybe you can add: Meniscal injury; Meniscal treatment; knee rehabilitation.
I see the valuable modification of your keywords, however, e.g., “knee” is very weak.

Introduction:
L34/35: Thanks, for the clarification of hoop tension. However, please, revise sentence L34/35 into two disjoint sentences or clearly write that increased contact pressure represents a consequence of a loss of hoop tension.

L53-55: Your hypotheses are still not clearly directed. You should at least clarify on hypotheses 1) “…would be affected by some certain pattern of affecting factors…”. I mean, what are your reasonable expectations after unfolding the theoretical content? What has to be verified and falsified in relation to affecting factors on MMPRT? Just stating that some things will change or will have an affect is not a valuable base for clearly manufactured scientific studies. You should have detected a clear literature driven lack of knowledge and / or research question, which will be falsified or verified by your analyses and results.

Methods:

Thanks for the valuable work on the Methods section.

L128: Delete “of”.

L125 et seq.: What post hoc analysis did you compute after finding significant differences computed by the ANOVA? I still cannot find any information about that. And I do not see any results results of pairwise analyses. The ANOVA reveals global differences between the dependent variables; but each significant result leads to the computation of pairwise post hoc t-tests to clarify which groups really significantly differ from each other. Please revise the Table 1 according to that. This is important information. By the significance of the ANOVA I just don’t know which groups initially differed between each other.

L130: Please write: Level of significance was set a priori at p<0.05.

Results:

Table 1: I still don’t get why and how you calculated statistical significance tests to the male/female ratio. What’s the rationale for that? Additionally, you commented that you computed post hoc analysis after ANOVA was significant. But I do not see any results of that computation. I want to clearly know, which groups (arthroplasty vs. HTO vs. Arthroscopic Tr. vs. Conservative Tr.) initially differ from each other. Just telling that you calculated post hoc is not sufficient.

L172: You’re reporting significant correlations of WBL ratio to delta WBL. I mean, aren’t these variables dependent from each other and wouldn’t we expect that they correlate with each other? What does the negative r-value stand for in that context? Give a practical explanation, please.

Discussion / Conclusion:

Good job, especially, at the beginning of the Discussion. The conclusion also fits to your results. Please, sum up your major conclusive findings in one sentence and add it at the end of the Abstract. Because therein this clear conclusion of your results is missing.
